# Choline supplementation: Impact on broiler chicken performance, steatosis, and economic viability from from 1 to 42 days

**Matheus Ramalho de Lima**[1]☯*, **Isabelle Naemi Kaneko**[2]‡, **Adiel Vieira de Lima**[2]‡, **Lucas Nunes de Melo**[2]‡, **Mario Cesar de Lima**[2]‡, **Anna Neusa Eduarda Ferreira de Brito**[2]‡, **Fernando Guilherme Perazzo Costa**[2]☯, **Andreia D. C. Vilas Boas**[3]☯, **Ana Louise Toledo**[3]☯, **Sigfrido Lopez Ferrer**[4]☯, **Saravanakumar Marimuthu**[5]☯

1 Federal University of Semi-Arid Region, Mossoró, Rio Grande do Norte, Brazil, 2 Federal University of Paraiba, Areia, Paraiba, Brazil, 3 Vidara do Brasil Ltda, Jundiai, São Paulo, Brazil, 4 Vidara Life Ingredients SAU, Barcelona, Espanha, 5 Natural Remedies Private Limited, Bangalore, India

☯ These authors contributed equally to this work.
‡ INK, AVL, LNM, MCL and ANEFB also contributed equally to this work.
* mrlmatheus@ufersa.edu.br

**Data Availability Statement:** All relevant data are within the manuscript and its Supporting Information files.

## Abstract

This study was carried out to compare the impact of choline supplementation (available from two sources synthetic and natural) on various dosages in broilers. The mode of choline supplementation, *via* diet and additional sources, synthetic and natural, and the data of performance, carcass quality, blood parameters, and hepatic steatosis were compared. A total of 1050 day-old male Cobb 500 broiler chicks were randomly assigned to 10 treatments, using a completely randomized design model in a factorial scheme, with 6 replicates per treatment and 25 birds per replicate. Choline was supplemented using three sources: synthetic choline chloride 60% (CC), and two sources of natural choline A (NCA), and B (NCB). The Control treatment did not receive any choline supplementation. The diets were supplemented with low, intermediate and high doses of choline sources (400g/t, 800g/t, and 1200g/t of CC; 100g/t, 200g/t, and 300g/t of both NCA and NCB). Data analysis was performed using a factorial model to investigate the effects of choline supplementation (CC, NCA, NCB) and doses on the measured variables. Overall, the results indicated that the the performance of NCA was better than CC & NCB, specifically the dose of 100g/t of NCA outperformed MAR at 100g/t & CC at 400g/t, leading to a significant increase in body weight gain (85.66g & 168.84g respectively), and a noteworthy (9- & 12-point respectively) improvement in feed conversion ratio. Furthermore, NCA contributed to a reduction in steatosis when contrasted with various NCB & CC doses, likely due to the presence of curcumins and catechins in the natural choline source. These findings demonstrated that NCA supplementation yielded superior results compared to CC and NCB across both performance and liver health aspects in broilers aged 1 to 42 days. In conclusion, NCA can be used to replace the CC 60% without compromise on the zootechnical performance in broilers.

**Funding:** The author(s) received no specific funding for this work.

## Introduction

The use of choline in commercial broiler feed is an important aspect in the feed industry to enhances the growth performance and regulates the lipid metabolism as well. Typically, choline plays a significant role in many important metabolic pathways, being crucial for the structural maintenance of cell membranes and organelles, acting as a constituent of phospholipids [1, 2] synthesis of methionine from homocysteine as a labile methyl donor; and a precursor molecule for the formation of acetylcholine in the nervous system. In addition, the choline is a component of very-low-density lipoproteins (VLDL) which involves in triglyceride transport out of the liver and thereby reduces the accumulation of fat in the liver [3–6]. It also inhibits the fatty acid synthesis by downregulating fatty acid synthase gene expression as well as attenuating its activity [7, 8]. Furthermore, choline is referred to as a "lipotropic" factor due to its role in increasing fat utilization, which results in the reduction of fat deposition in the body [5, 9].

However, the absence of choline, or levels below the recommendation, increases deleterious effects on meat chickens, such as reduction in growth and perosis, especially in younger birds [5, 10]. Choline supplementation is commonly done by synthetic sources, but high hygroscopicity and oxidation with loss of vitamins, its corrosive nature, as well as the trimethylamine formation in the intestinal tract of broilers (the common reason less than half of the choline chloride is absorbed), allowed th te scientist to look for an alternative [11, 12].

In this respect, research has been developed to unveil potential substitutes for synthetic sources, and formulations with herbs have been gaining ground with consistent results in hepatoprotective capacity and improvement of performance parameters. These natural products, produced from selected plants and blends of herbs, which can mimic the function of choline [12]. Chandrasekaran et. al [13] when investigating the lipotropic activity of herbal formulations containing *Acacia nilotica* and *Curcuma longa*, demonstrated the lipotropic effect of this natural source which was highlighted by low fat accumulation and anti-lipogenic activity. The formulations of natural choline using herbs such as *Acacia nilotica* and *Curcuma longa* paved a way to get the associating benefits and thus enhancing the effects of hepatoprotective action which was being reported discretely [14, 15].

Based on these infromations, it is possible to say the natural choline supplementation, specifically with *Acacia nilotica and Curcuma longa*, will yield superior results compared to synthetic choline supplementation. This superiority will be evidenced by improved performance parameters, reduced hepatic steatosis, and enhanced hepatoprotective action. This hypothesis is based on the premise that natural sources of choline not only mimic the function of synthetic choline but also offer additional benefits such as hepatoprotective action. Hence the current study was carried out to compare the impact of choline supplementation (available from two sources synthetic source and natural sources) on various dosages in broilers. The mode of choline supplementation, *via* diet and additional sources, synthetic and natural, and the data of performance, carcass quality, blood parameters, and hepatic steatosis were compared.

## Material and methods

### Housing conditions and management

Housing, feeding regimens, and rearing conditions used at the Federal University of Paraiba (UFPB) are thought to be representative of modern commercial broiler operations in Brazil. This trial was carried out at an experimental farm located in Areia–Paraiba, Brazil (6 degrees, 57 minutes, and 48 seconds south of the equator (latitude), 35 degrees, 41 minutes, and 30 seconds west of the prime meridian (longitude), and 618 meters above sea level (altitude).

Farm facilities comprise 88 identical floor pens (2.25m$^2$). Of these, 42 pens were used. The trial was carried out with 1050 day-old male Cobb 500 broiler chicks, 39.66 ± 0.20g, organized

in a factorial experiment, 3 x 3 + 1. The effect size was calculated using Cohen's d. A total of 1050 day-old male Ross 500 broiler chicks were randomly assigned to 10 treatments, using a completely randomized design model in a factorial scheme, with 6 replicates per treatment and 25 birds per replicate. Individual pens measuring 1.5 x 1.5 m served as experimental units. Pens were consecutively numbered with the respective treatment number using pen cards.

Birds were kept in appropriate (i.e., optimum) environmental conditions (temperature, from ±32˚C from 1-7d; ±30˚C from 8-14d; ±26˚C from 15 to 42d) for age during the trial. Lighting was provided using fluorescent bulbs, To maintain an intensity of 23 lux, with 24 hours for 7 days, and 23 hours of light up to 42 days. The experimental period totalled 42 days. A 3-phase feeding program was used as follows: starter (days 1–21), grower (days 22–35), and finisher (days 36–42). Feed was provided in a pan feeder (capacity, 20 kg). Water was offered *ad libitum* using drinkers.

Standard floor pen management practices were used throughout the experimental period. Birds and facilities were inspected twice daily. The following data were recorded: general health status of broilers, environmental temperature, and constant feed and water supply. Dead birds were removed, and unexpected events were identified. Deaths, including pen number, date of death, bird weight, and potential diagnosis, were recorded using a Daily Mortality Record.

Day-old male Cobb chicks hatched from eggs produced by 30 weeks-old breeders and weighing 40g on average were obtained from Frango Dourado hatchery (Pernambuco, Brazil). Twenty-five male broiler chicks were housed in each pen. Birds were sexed at the hatchery by feather sexing method.

### Dietary treatments

Choline was supplemented using three sources: synthetic choline chloride 60% (CC), and two sources of natural choline A (NCA), and B (NCB). The Control treatment did not receive any choline supplementation. The diets were supplemented with low, intermediate and high doses of choline sources (400g/t, 800g/t, and 1200g/t of CC; 100g/t, 200g/t, and 300g/t of both NCA and NCB). Natural Choline A (NCA) is indexed as Kolin Plus (M/s Natural Remedies Pvt Ltd, Bengaluru, India), is a polyherbal formulation containing a combination of *Acacia nilotica* (*A. nilotica*) and *Curcuma longa* (C. longa) belonging to the families of Mimosaceae and Zingiberaceae, respectively. Natural Choline B (NCB) contains mainly *Achyranthes aspera*, *Trachyspermum ammi*, *Azadirachta indica*, and others.

### Feed

Mash feeds were manufactured at UFPB Agrarian Center—Campus II, located in Areia, Paraiba, Brazil. Broiler diets were formulated with feedstuffs widely used in Brazil. Diets were representative of local commercial feed formulations and designed to meet or exceeded nutritional requirements for broiler strain and age. Feed batches were independently mixed and bagged. Feed bags were labeled with a trial number, mixing date, type of feed, dietary treatment, and replicate number. Detailed records of feed mixtures and test product inventories were kept. Test feeds were mixed for 4 minutes at 28 rev/min in a 100 kg capacity mixer. The experimental diets were described in Tables 1–3.

### Measurements

**Performance.** In this study, the daily mortality rate of the chicks was recorded. The body weight (BW, g/broiler) of each chick was measured on a scale on Day 0, Day 21, Day 35, and Day 42 (end of the experimental period), and these measurements were used to calculate the

**Table 1. Experimental diets from 1 to 21 days.**

| | Investment | Basal | CC | CC | CC | NCA | NCA | NCA | NCB | NCB | NCB |
|---|---|---|---|---|---|---|---|---|---|---|---|
| | kg | | 400 g/t | 800 g/t | 1,200 g/t | 100 g/t | 200 g/t | 300 g/t | 100 g/t | 200 g/t | 300 g/t |
| **Macro Ingredients** | | | | | | | | | | | |
| Corn 7.8% | BRL 1.50 | 575.1 | 574.69 | 574.29 | 572.89 | 575 | 574.9 | 574.79 | 575 | 574.9 | 574.79 |
| Soybean meal 45% | BRL 3.00 | 334 | 334 | 334 | 334 | 334 | 334 | 334 | 334 | 334 | 334 |
| Meat meal 38% | BRL 3.00 | 29 | 29 | 29 | 29 | 29 | 29 | 29 | 29 | 29 | 29 |
| Degummed soybean oil | BRL 6.00 | 37 | 37 | 37 | 38 | 37 | 37 | 37 | 37 | 37 | 37 |
| Limestone 36% | BRL 0.25 | 8.2 | 8.2 | 8.2 | 8.2 | 8.2 | 8.2 | 8.2 | 8.2 | 8.2 | 8.2 |
| Common salt | BRL 0.60 | 4.6 | 4.6 | 4.6 | 4.6 | 4.6 | 4.6 | 4.6 | 4.6 | 4.6 | 4.6 |
| Mineral and vitamin premix 0.4%[1] | BRL 1.00 | 4 | 4 | 4 | 4 | 4 | 4 | 4 | 4 | 4 | 4 |
| **Micro Ingredients** | | | | | | | | | | | |
| Choline chloride 60%, CC | BRL 13.68 | | 0.4 | 0.8 | 1.2 | | | | | | |
| Natural Choline, NCA | BRL 27.35 | | | | | 0.1 | 0.2 | 0.3 | | | |
| Natural Choline, NCB | BRL 32.82 | | | | | | | | 0.1 | 0.2 | 0.3 |
| L-Lysine HCL | BRL 8.00 | 3.15 | 3.15 | 3.15 | 3.15 | 3.15 | 3.15 | 3.15 | 3.15 | 3.15 | 3.15 |
| DL-Methionine 99% | BRL 18.00 | 3.7 | 3.7 | 3.7 | 3.7 | 3.7 | 3.7 | 3.7 | 3.7 | 3.7 | 3.7 |
| L-Threonine | BRL 7.00 | 1.25 | 1.26 | 1.26 | 1.26 | 1.25 | 1.25 | 1.26 | 1.25 | 1.25 | 1.26 |
| **Total Mixed** (kg) | | **1,000.00** | **1,000.00** | **1,000.00** | **1,000.00** | **1,000.00** | **1,000.00** | **1,000.00** | **1,000.00** | **1,000,00** | **1,000,00** |
| **End Price** (BRL/kg) | | BRL 2.28 | BRL 2.29 | BRL 2.29 | BRL 2.29 | BRL 2.28 | BRL 2.28 | BRL 2.29 | BRL 2.29 | BRL 2.29 | BRL 2.30 |
| Crude protein | % | 21.22 | 21.22 | 21.22 | 21.22 | 21.22 | 21.22 | 21.22 | 21.22 | 21.22 | 21.22 |
| Crude fiber | % | 3.35 | 3.35 | 3.35 | 3.35 | 3.35 | 3.35 | 3.35 | 3.35 | 3.35 | 3.35 |
| Calcium | % | 0.99 | 0.99 | 0.99 | 0.99 | 0.99 | 0.99 | 0.99 | 0.99 | 0.99 | 0.99 |
| Available phosphorus | % | 0.47 | 0.47 | 0.47 | 0.47 | 0.47 | 0.47 | 0.47 | 0.47 | 0.47 | 0.47 |
| Metabolizable energy | kcal/kg | 3.140.00 | 3.140.00 | 3.140.00 | 3.140.00 | 3.140.00 | 3.140.00 | 3.140.00 | 3.140.00 | 3.140.00 | 3.140.00 |
| Digestible lysine | % | 1.26 | 1.26 | 1.26 | 1.26 | 1.26 | 1.26 | 1.26 | 1.26 | 1.26 | 1.26 |
| Digestible methionine | % | 0.65 | 0.65 | 0.65 | 0.65 | 0.65 | 0.65 | 0.65 | 0.65 | 0.65 | 0.65 |
| Digestible methionine + cystine | % | 0.93 | 0.93 | 0.93 | 0.93 | 0.93 | 0.93 | 0.93 | 0.93 | 0.93 | 0.93 |
| Digestible threonine | % | 0.83 | 0.83 | 0.83 | 0.83 | 0.83 | 0.83 | 0.83 | 0.83 | 0.83 | 0.83 |
| Choline | mg/kg | 1.162.65 | 1.162.49 | 1.162.54 | 1.162.60 | 1.162.60 | 1.162.54 | 1.162.49 | 1.162.60 | 1.162.54 | 1.162.49 |
| Phosphatidylcholine | g/kg | - | - | - | - | 25 | 50 | 75 | 16 | 32 | 48 |
| Sodium | % | 0.22 | 0.22 | 0.22 | 0.22 | 0.22 | 0.22 | 0.22 | 0.22 | 0.22 | 0.22 |
| Chlorine | % | 0.41 | 0.41 | 0.41 | 0.41 | 0.41 | 0.41 | 0.41 | 0.41 | 0.41 | 0.41 |

Dollar equivalent to BRL 5.47, July 20, 2022. [1]Minimum premix per kilogram of feed: Mn: 60 g; Fe: 80 g; Zn: 50 g; Cu: 10 g; Co: 2 g; I: 1 g; Se: 250 mg.; Vitamin per kilogram of feed: vitamin A: 15,000,000 IU; vitamin D3: 1,500,000 IU; vitamin E: 15,000 IU; vitamin B1: 2.0 g; vitamin B2: 4.0 g; vitamin B6: 3.0 g; vitamin B12: 0.015 g; nicotinic acid: 25 g; pantothenic acid: 10 g; vitamin K3: 3.0 g; folic acid: 1.0 g.

body weight gain (BWG, g/broiler). The feed intake (FI, g/broiler) was determined by measuring and recording the amount of feed consumed by the chicks on Day 21, Day 35, and Day 42. The difference between the initial and final weights for each period provided the amount of feed intake. The body weight gain (BWG) was calculated by subtracting the initial body weight (on Day 0) from the final body weight (on Day 42). The feed conversion ratio (FCR, g/g) was then calculated by dividing the total feed intake by the total body weight gain. This calculation was adjusted for mortality, meaning the weight of the chicks that died was subtracted from the total body weight gain before the FCR was calculated.

**Carcass.** On Day 42, after performance data collection, 5 broilers from each replicate, selected by average weight, were euthanized for assessment of various traits. The carcass characteristics were evaluated, providing insight into the physical and qualitative attributes of the

**Table 2. Experimental diets from 22 to 35 days.**

| | Investment | Basal | CC | CC | CC | NCA | NCA | NCA | NCB | NCB | NCB |
|---|---|---|---|---|---|---|---|---|---|---|---|
| | kg | | 400 g/t | 800 g/t | 1,200 g/t | 100 g/t | 200 g/t | 300 g/t | 100 g/t | 200 g/t | 300 g/t |
| **Macro Ingredients** | | | | | | | | | | | |
| Corn 7.8% | BRL 1.50 | 604.29 | 603.88 | 602.48 | 602.08 | 604.19 | 604.09 | 603.98 | 604.19 | 604.09 | 603.98 |
| Soybean meal 45% | BRL 3.00 | 299 | 299 | 299 | 299 | 299 | 299 | 299 | 299 | 299 | 299 |
| Meat meal 38% | BRL 3.00 | 24 | 24 | 25 | 25 | 24 | 24 | 24 | 24 | 24 | 24 |
| Degummed soybean oil | BRL 6.00 | 49 | 49 | 49 | 49 | 49 | 49 | 49 | 49 | 49 | 49 |
| Limestone 36% | BRL 0.25 | 8.3 | 8.3 | 8.3 | 8.3 | 8.3 | 8.3 | 8.3 | 8.3 | 8.3 | 8.3 |
| Common salt | BRL 0.60 | 4.2 | 4.2 | 4.2 | 4.2 | 4.2 | 4.2 | 4.2 | 4.2 | 4.2 | 4.2 |
| Mineral and vitamin premix 0.4%[1] | BRL 1.00 | 4 | 4 | 4 | 4 | 4 | 4 | 4 | 4 | 4 | 4 |
| **Micro Ingredients** | | | | | | | | | | | |
| Choline chloride 60%, CC | BRL 13.68 | | 0.4 | 0.8 | 1.2 | | | | | | |
| Natural Choline, NCA | BRL 27.35 | | | | | 0.1 | 0.2 | 0.3 | | | |
| Natural Choline, NCB | BRL 32.82 | | | | | | | | 0.1 | 0.2 | 0.3 |
| L-Lysine HCL | BRL 8.00 | 2.93 | 2.93 | 2.93 | 2.93 | 2.93 | 2.93 | 2.93 | 2.93 | 2.93 | 2.93 |
| DL-Methionine 99% | BRL 18.00 | 3.23 | 3.24 | 3.24 | 3.24 | 3.23 | 3.23 | 3.24 | 3.23 | 3.23 | 3.24 |
| L-Threonine | BRL 7.00 | 1.05 | 1.05 | 1.05 | 1.05 | 1.05 | 1.05 | 1.05 | 1.05 | 1.05 | 1.05 |
| **Total Mixed** (kg) | | 1,000.00 | 1,000.00 | 1,000.00 | 1,000.00 | 1,000.00 | 1,000.00 | 1,000.00 | 1,000.00 | 1,000.00 | 1,000.00 |
| **End Price** (BRL/kg) | | BRL 2.27 | BRL 2.28 | BRL 2.27 | BRL 2.27 | BRL 2.27 | BRL 2.27 | BRL 2.27 | BRL 2.27 | BRL 2.28 | BRL 2.28 |
| Crude protein | % | 19.64 | 19.63 | 19.63 | 19.63 | 19.64 | 19.63 | 19.63 | 19.64 | 19.63 | 19.63 |
| Crude fiber | % | 3.18 | 3.17 | 3.17 | 3.17 | 3.18 | 3.18 | 3.18 | 3.18 | 3.17 | 3.17 |
| Calcium | % | 0.92 | 0.92 | 0.92 | 0.92 | 0.92 | 0.92 | 0.92 | 0.92 | 0.92 | 0.92 |
| Available phosphorus | % | 0.43 | 0.43 | 0.43 | 0.43 | 0.43 | 0.43 | 0.43 | 0.43 | 0.43 | 0.43 |
| Metabolizable energy | kcal/kg | 3,250.00 | 3,250.00 | 3,250.00 | 3,250.00 | 3,250.00 | 3,250.00 | 3,250.00 | 3,250.00 | 3,250.00 | 3,250.00 |
| Digestible lysine | % | 1.15 | 1.15 | 1.15 | 1.15 | 1.15 | 1.15 | 1.15 | 1.15 | 1.15 | 1.15 |
| Digestible methionine | % | 0.58 | 0.58 | 0.58 | 0.58 | 0.58 | 0.58 | 0.58 | 0.58 | 0.58 | 0.58 |
| Digestible methionine + cystine | % | 0.85 | 0.85 | 0.85 | 0.85 | 0.85 | 0.85 | 0.85 | 0.85 | 0.85 | 0.85 |
| Digestible threonine | % | 0.76 | 0.76 | 0.76 | 0.76 | 0.76 | 0.76 | 0.76 | 0.76 | 0.76 | 0.76 |
| Choline | mg/kg | 1,083.45 | 1,262.05 | 1,440.65 | 1,619.24 | 1,083.40 | 1,083.35 | 1,083.30 | 1,083.40 | 1,083.35 | 1,083.30 |
| Phosphatidyl choline | g/kg | | | | | 25 | 50 | 75 | 16 | 32 | 48 |
| Sodium | % | 0.2 | 0.2 | 0.2 | 0.2 | 0.2 | 0.2 | 0.2 | 0.2 | 0.2 | 0.2 |
| Chlorine | % | 0.37 | 0.37 | 0.37 | 0.37 | 0.37 | 0.37 | 0.37 | 0.37 | 0.37 | 0.37 |

Dollar equivalent to BRL 5.47, July 20, 2022. [1]Minimum premix per kilogram of feed: Mn: 60 g; Fe: 80 g; Zn: 50 g; Cu: 10 g; Co: 2 g; I: 1 g; Se: 250 mg.; Vitamin per kilogram of feed: vitamin A: 15,000,000 IU; vitamin D3: 1,500,000 IU; vitamin E: 15,000 IU; vitamin B1: 2.0 g; vitamin B2: 4.0 g; vitamin B6: 3.0 g; vitamin B12: 0.015 g; nicotinic acid: 25 g; pantothenic acid: 10 g; vitamin K3: 3.0 g; folic acid: 1.0 g.

broiler carcasses. The amount of abdominal fat, an indicator of the overall health and condition of the broilers, was determined. The yields of the whole carcass, breast meat, thigh meat, and drumstick meat were also determined. These yields, calculated as a percentage of the broiler's total body weight.

**Fatty liver data.** On D42, liver samples (1 cm$^2$) were collected and coated with talc, immediately frozen in liquid nitrogen, and stored in an ultra-freezer. Frozen samples were cut into 20 μm thick sections using a cryostat-microtome for historical slide preparation. Slides were then stained with Sudan IV for lipid quantification. Areas of lipid deposition were stained in red. Samples were scored as follows: 1—no lipid quantification, 2—few areas of lipid quantification, 3—larger areas of lipid quantification, and 4—lipid quantification over all tissue.

**Economic analysis.** Combined analysis of liveability (i.e., the final number of birds per treatment), performance, carcass yield, and economic data, which can be adjusted according

**Table 3. Experimental diets from 36 to 42 days.**

| | Investment | Basal | CC | CC | CC | NCA | NCA | NCA | NCB | NCB | NCB |
|---|---|---|---|---|---|---|---|---|---|---|---|
| | kg | | 400 g/t | 800 g/t | 1,200 g/t | 100 g/t | 200 g/t | 300 g/t | 100 g/t | 200 g/t | 300 g/t |
| **Macro Ingredients** | | | | | | | | | | | |
| Corn 7.8% | BRL 1.50 | 639.33 | 637.93 | 637.53 | 637.12 | 638.24 | 638.14 | 638.03 | 638.24 | 638.14 | 638.03 |
| Soybean meal 45% | BRL 3.00 | 264 | 265 | 265 | 265 | 265 | 265 | 265 | 265 | 265 | 265 |
| Meat meal 38% | BRL 3.00 | 19 | 19 | 19 | 19 | 19 | 19 | 19 | 19 | 19 | 19 |
| Degummed soybean oil | BRL 6.00 | 55 | 55 | 55 | 55 | 55 | 55 | 55 | 55 | 55 | 55 |
| Limestone 36% | BRL 0.25 | 8.5 | 8.5 | 8.5 | 8.5 | 8.5 | 8.5 | 8.5 | 8.5 | 8.5 | 8.5 |
| Common salt | BRL 0.60 | 3.8 | 3.8 | 3.8 | 3.8 | 3.8 | 3.8 | 3.8 | 3.8 | 3.8 | 3.8 |
| Mineral and vitamin premix 0.4%[1] | BRL 1.00 | 4 | 4 | 4 | 4 | 4 | 4 | 4 | 4 | 4 | 4 |
| **Micro Ingredients** | | | | | | | | | | | |
| Choline chloride 60%, CC | BRL 13.68 | | 0.4 | 0.8 | 1.2 | | | | | | |
| Natural Choline, NCA | BRL 27.35 | | | | | 0.1 | 0.2 | 0.3 | | | |
| Natural Choline, NCB | BRL 32.82 | | | | | | | | 0.1 | 0.2 | 0.3 |
| L-Lysine HCL | BRL 8.00 | 2.75 | 2.74 | 2.74 | 2.74 | 2.74 | 2.74 | 2.74 | 2.74 | 2.74 | 2.74 |
| DL-Methionine 99% | BRL 18.00 | 2.77 | 2.78 | 2.78 | 2.78 | 2.77 | 2.77 | 2.78 | 2.77 | 2.77 | 2.78 |
| L-Threonine | BRL 7.00 | 0.85 | 0.85 | 0.85 | 0.86 | 0.85 | 0.85 | 0.85 | 0.85 | 0.85 | 0.85 |
| **Total Mixed** (kg) | | **1,000.00** | **1,000.00** | **1,000.00** | **1,000.00** | **1,000.00** | **1,000.00** | **1,000.00** | **1,000.00** | **1,000.00** | **1,000.00** |
| **End Price** (BRL/kg) | | BRL 2.22 | BRL 2.23 | BRL 2.23 | BRL 2.23 | BRL 2.23 | BRL 2.23 | BRL 2.23 | BRL 2.23 | BRL 2.23 | BRL 2.24 |
| Crude protein | % | 18.09 | 18.08 | 18.08 | 18.08 | 18.09 | 18.09 | 18.08 | 18.09 | 18.09 | 18.08 |
| Crude fiber | % | 3.02 | 3.02 | 3.01 | 3.01 | 3.02 | 3.02 | 3.02 | 3.02 | 3.02 | 3.02 |
| Calcium | % | 0.83 | 0.83 | 0.83 | 0.83 | 0.83 | 0.83 | 0.83 | 0.83 | 0.83 | 0.83 |
| Available phosphorus | % | 0.39 | 0.39 | 0.39 | 0.39 | 0.39 | 0.39 | 0.39 | 0.39 | 0.39 | 0.39 |
| Metabolizable energy | kcal/kg | 3,330.00 | 3,330.00 | 3,330.00 | 3,330.00 | 3,330.00 | 3,330.00 | 3,330.00 | 3,330.00 | 3,330.00 | 3,330.00 |
| Digestible lysine | % | 1.04 | 1.04 | 1.04 | 1.04 | 1.04 | 1.04 | 1.04 | 1.04 | 1.04 | 1.04 |
| Digestible methionine | % | 0.52 | 0.52 | 0.52 | 0.52 | 0.52 | 0.52 | 0.52 | 0.52 | 0.52 | 0.52 |
| Digestible methionine + cystine | % | 0.77 | 0.77 | 0.77 | 0.77 | 0.77 | 0.77 | 0.77 | 0.77 | 0.77 | 0.77 |
| Digestible threonine | % | 0.69 | 0.69 | 0.69 | 0.69 | 0.69 | 0.69 | 0.69 | 0.69 | 0.69 | 0.69 |
| Choline | mg/kg | 1,007.18 | 1,185.78 | 1,364.38 | 1,542.98 | 1,007.13 | 1,007.08 | 1,007.03 | 1,007.13 | 1,007.08 | 1,007.03 |
| Phosphatidyl choline | g/kg | | | | | 25 | 50 | 75 | 16 | 32 | 48 |
| Sodium | % | 0.18 | 0.18 | 0.18 | 0.18 | 0.18 | 0.18 | 0.18 | 0.18 | 0.18 | 0.18 |
| Chlorine | % | 0.34 | 0.34 | 0.34 | 0.34 | 0.34 | 0.34 | 0.34 | 0.34 | 0.34 | 0.34 |

Dollar equivalent to BRL 5.47, July 20, 2022. [1]Minimum premix per kilogram of feed: Mn: 60 g; Fe: 80 g; Zn: 50 g; Cu: 10 g; Co: 2 g; I: 1 g; Se: 250 mg.; Vitamin per kilogram of feed: vitamin A: 15,000,000 IU; vitamin D3: 1,500,000 IU; vitamin E: 15,000 IU; vitamin B1: 2.0 g; vitamin B2: 4.0 g; vitamin B6: 3.0 g; vitamin B12: 0.015 g; nicotinic acid: 25 g; pantothenic acid: 10 g; vitamin K3: 3.0 g; folic acid: 1.0g.

to natural variations, revealed relative gross margin fluctuations across different scenarios. Using the Control treatment as a reference, the relative gross income derived from breast, thigh, drumstick, and whole chicken sales is 100%. The same applies to the remaining treatments in the same order. Hence, 100 can be used as a benchmark to estimate the difference between treatments.

**Liver cholesterol, triglycerides, and lipids.** On D42, liver samples (1 cm$^2$) were collected. One set (Triplicates of ~1 g of tissue) of samples were collected from dispersed zones of the frozen left lobe. The exact wet weight of each sample was determined after thawing and dehydrating the excess moisture on a Whatman filter paper for 10 min at 25°C. Then the total fat was extracted from the liver samples followed by liver triglycerides and cholesterol were quantified using the [16], and lipids content using [17].

**Statistical analysis.** The statistical analysis of the characteristics studied was performed using the R software, version 4.2.1. The analysis of variance (ANOVA) assumptions (error normality, random and independent errors, and variance homoscedasticity) was met.

Data analysis was performed using a factorial model to investigate the effects of choline supplementation (CC, NCA, NCB) and doses on the measured variables. The factorial model can be represented as $Y_{ijk} = \mu + \alpha_i + \beta_j + (\alpha\beta)_{ij} + \varepsilon_{ijk}$, where: $Y_{ijk}$ is the dependent variable (the measured variables in the study), $\mu$ is the overall mean, $\alpha_i$ is the effect of the level of choline supplementation (CC, NCA, NCB), $\beta_j$ is the effect of the level of dose, $(\alpha\beta)_{ij}$ is the interaction effect between choline supplementation and dose, and $\varepsilon_{ijk}$ is the random error term.

A two-way analysis of variance (ANOVA) was conducted to assess the main effects of choline supply and doses, as well as any interactions between them. The significance level was set at $\alpha = 0.05$ for all tests. Post hoc Tukey tests were conducted to compare the means of each choline source and dose to identify specific differences.

## Results

There was a significant interaction observed in body weight (BW) (P = 0.0405), body weight gain (BWG) (P = 0.0394), and feed intake (FI) (P = 0.0139), as shown in Table 4. In the overall effect, birds exhibited higher weights with CC supplementation compared to the Without group and NCB (P = 0.033), but similar to NCA (P>0.05). Weight gain significantly improved with the intermediate dosage compared to the control (P = 0.164), but was similar to the low and high dosages (P>0.05). The BWG data were consistent with the findings regarding final body weight in all aspects. There was no significant effect observed on feed consumption, feed conversion ratio, and survival rate.

Higher body weight (BW) was observed with choline supplementation via CC at the intermediate dosage (P = 0.034) and high dosage (P = 0.0032), while no significant difference was found at the low dosage (P = 0.1232), as evidenced by the data presented in Table 5. Notably, there was no observable effect on the incremental doses with CC supplementation (P = 0.0948)

**Table 4. Effect of choline supplementation on zootechnical parameters of broilers at 21 days of age.**

| Supplementation | IBW | BW | BWG | FI | FCR | LIVE % |
|---|---|---|---|---|---|---|
| Without | 39.605 | 866.131b | 826.525b | 1320.004 | 1.596 | 99.123 |
| CC | 39.626 | 894.498a | 854.872a | 1357.403 | 1.588 | 99.415 |
| NCA | 39.661 | 880.030ab | 840.370ab | 1304.835 | 1.553 | 98.538 |
| NCB | 39.716 | 870.052b | 830.335b | 1336.707 | 1.610 | 99.415 |
| **Dosage[1]** | **IBW** | **BW** | **BWG** | **FI** | **FCR** | **LIVE** |
| Control | 39.605 | 866.131b | 826.525b | 1320.004 | 1.596 | 99.123 |
| Low | 39.649 | 874.488ab | 834.839ab | 1335.071 | 1.600 | 99.123 |
| Intermediate | 39.675 | 892.914a | 853.239a | 1353.100 | 1.586 | 98.538 |
| High | 39.678 | 877.178ab | 837.500ab | 1310.773 | 1.565 | 99.708 |
| **Source** | **0.5521** | **0.0033** | **0.0031** | **0.3046** | **0.3764** | **0.4819** |
| **Dosage** | **0.88** | **0.0164** | **0.0162** | **0.4921** | **0.7422** | **0.3534** |
| **Interaction** | **1.000** | **0.0405** | **0.0394** | **0.0139** | **0.1377** | **0.097** |
| **SEM** | **0.05** | **4.7622** | **4.7572** | **19.5552** | **0.02424** | **0.4623** |
| **C.V. (%)** | **0.54** | **2.36** | **2.46** | **6.29** | **6.22** | **1.94** |

IBW–Initial body weight; BW–Body weight; BWG–Body weight gain; FI–Feed intake; FCR–Feed conversion ratio; LIVE–Liveability; [1]Dosage means the dose of each choline source, being Low (100g/t for NCA and NCB and 400g/t for CC), Intermediate (200g/t for NCA and NCB and 800g/t for CC), and High (300g/t for NCA and NCB and 1200g/t for CC). a,b Means followed by different letters in the same column differ significantly Tukey test, P<0.05.

**Table 5. Effect of interaction between the supplementation sources of broilers at 21 days of age.**

| BW | Control | CC | NCA | NCB | P value |
|---|---|---|---|---|---|
| Control | 866.131 | | | | |
| Low | | 880.526 | 882.831 | 860.106B | 0.1232 |
| Intermediate | | 907.010a | 875.366b | 896.365abA | 0.034 |
| High | | 895.957a | 881.893ab | 853.683bB | 0.0032 |
| P value | | 0.0948 | 0.7947 | 0.0015 | |
| SEM | | 5.5422 | 4.4332 | 3.2703 | |
| **BWG** | **Control** | **CC** | **NCA** | **NCB** | **P value** |
| Control | 826.525 | | | | |
| Low | | 840.921 | 843.182 | 820.413B | 0.1232 |
| Intermediate | | 867.378a | 835.699b | 856.637abA | 0.0333 |
| High | | 856.316a | 842.226ab | 813.955bB | 0.003 |
| P value | - | 0.094 | 0.793 | 0.0015 | |
| SEM | | 4.7253 | 4.2941 | 3.8713 | |
| **FI** | **Control** | **CC** | **NCA** | **NCB** | **P value** |
| Control | 1320.000 | | | | |
| Low | | 1356.491 | 1309.424 | 1339.298AB | 0.6188 |
| Intermediate | | 1357.958ab | 1278.326b | 1423.014aA | 0.0161 |
| High | | 1357.758 | 1326.754 | 1247.807B | 0.0739 |
| P value | | 0.9995 | 0.601 | 0.0029 | |
| SEM | | 20.7263 | 19.1435 | 18.1241 | |

BW–Body weight; BWG–Body weight gain; FI–Feed intake; a,b Means followed by different letters in the same line differ significantly Tukey test, P<0.05. A, B Means followed by different letters in the same column differ significantly Tukey test, P<0.05.

or NCA supplementation (P = 0.7947). However, NCB supplementation resulted in a significantly higher final weight at the intermediate dosage compared to the low and high dosages (P = 0.0150). The BWG data exhibited similar patterns to those observed in BW. Regarding feed intake, significantly higher values were obtained with NCB supplementation compared to NCA (P = 0.0161), although no significant difference was observed between NCB and CC (P>0.05). Moreover, NCB showed greater feed intake at the intermediate dosage compared to the high dosage (P = 0.0029), while no significant difference was found between the intermediate and low dosages (P>0.05).

There was a significant interaction observed in BW (P = 0.0164), BWG (P = 0.0164), FI (P = 0.0339), and FCR (P = 0.0097), as presented in Table 6. In the overall effect, feed conversion ratio (FCR) showed higher values with NCB supplementation compared to NCA (P = 0.008) but was similar to CC and the Without supplementation (P>0.05).

Higher BW was observed with choline supplementation *via* CC at intermediate and high dosages compared to low dosages (P = 0.099), as indicated by the data presented in Table 7. The BWG data showed similar effects to those obtained in BW. NCB exhibited higher feed intake at the intermediate dosage compared to the high dosage (P = 0.0039) but was similar to the low dosage (P>0.05). The FCR demonstrated better results with NCA compared to CC and NCB at the low dosage (P = 0.0425). At the intermediate dosage, the FCR showed higher values with the NCB source compared to the CC and NCA sources (P = 0.003). The FCR with the use of CC (P = 0.2294) or NCA (P = 0.5825) is similar to the evaluated dosages, whereas, with NCB, there is a decrease in effectiveness at the intermediate dosage compared to the other dosages of this additive (P = 0.0026).

**Table 6. Effect of choline supplementation on zootechnical parameters of broilers at 35 days of age.**

| Supplementation | BW | BWG | FI | FCR | LIVE |
|---|---|---|---|---|---|
| Without | 2147.639 | 2108.034 | 3586.367 | 1.704ab | 96.491 |
| CC | 2199.488 | 2159.862 | 3648.010 | 1.690ab | 96.491 |
| NCA | 2184.211 | 2144.550 | 3560.725 | 1.661b | 97.076 |
| NCB | 2154.361 | 2114.645 | 3671.365 | 1.736a | 97.368 |
| **Dosage[1]** | **BW** | **BWG** | **FI** | **FCR** | **LIVE** |
| Control | 2147.639 | 2108.034 | 3586.367 | 1.704 | 96.491 |
| Low | 2167.512 | 2127.863 | 3606.538 | 1.696 | 96.491 |
| Intermediate | 2188.708 | 2149.032 | 3690.273 | 1.718 | 96.491 |
| High | 2181.841 | 2142.162 | 3583.288 | 1.673 | 97.953 |
| **Source** | **0.2937** | **0.2929** | **0.1809** | **0.008** | **0.8704** |
| **Dosage** | **0.6818** | **0.6831** | **0.2054** | **0.2035** | **0.5254** |
| **Interaction** | **0.0164** | **0.0164** | **0.0339** | **0.0097** | **0.2538** |
| **SEM** | **12.6839** | **12.6845** | **41.7221** | **0.0018** | **1.0065** |
| **C.V. (%)** | **3.67** | **3.74** | **4.44** | **3.73** | **3.56** |

IBW–Initial body weight; BW–Body weight; BWG–Body weight gain; FI–Feed intake; FCR–Feed conversion ratio; LIVE–Liveability; [1]Dosage means the dose of each choline source, being Low (100g/t for NCA and NCB and 400g/t for CC), Intermediate (200g/t for NCA and NCB and 800g/t for CC), and High (300g/t for NCA and NCB and 1200g/t for CC). a,b Means followed by different letters in the same column differ significantly Tukey test, P<0.05.

There was a significant interaction observed in BW (P = 0.0166), BWG (P = 0.0167), and FCR (P = 0.0457), as shown in Table 8. In the overall effect, FCR showed better results with NCA compared to NCB and CC (P = 0.0432), and was similar to the Without supplementation (P>0.05).

Higher BW was observed with choline supplementation *via* NCA at low dosages compared to CC and NCB (P = 0.0443), as shown in Table 9. The CC supplementation resulted in higher BW with intermediate and high doses compared to the low dose (P = 0.326), while NCA showed better results with the low dose compared to the others (P = 0.0203). The NCB was not influenced by the doses (P = 0.6306). Low dose weight gain was higher with NCA than with CC (P = 0.0445), but similar to NCB. FCR showed better results with NCA compared to NCB (P = 0.0405), but similar to CC.

NCA demonstrates similarity to NCB, however, FCR significantly differs between them in the 1–42 day phase (P = 0.0405). The similarity of NCA with CC in this variable indicates that they are equivalent in providing choline satisfactorily. However, NCA shows better results with doses of 100g/t to 200g/t, while CC shows better results with doses of 800g/t to 1200g/t.

The breast yield was lower when NCB was used (P = 0.0036), as shown in Table 10. On the other hand, this source promoted higher thigh yield (P = 0.0251). The leg yield was better with the low dose of choline supplementation (P = 0.0065).

The breast yield was influenced by the supplementation of the high dosage, with a superiority (P = 0.0002) of CC and NCA sources compared to NCB, Table 11. Regarding thighdrums yield, there was a better performance with NCB compared to the others (P = 0.0383). NCB performed better with the low dosage than the intermediate (P = 0.0017) but was similar to the high in ThighDrums yield.

A higher cholesterol content was obtained with the control group compared to the intermediate (P = 0.0591), while it was similar in the low and high choline supplementation dosages, as shown in Table 12. Meanwhile, the low dosage promoted a higher liver lipid content compared to the other dosages (P = 0.0394). In this context, the supplemented choline sources

**Table 7. Effect of interaction between the supplementation sources of broilers at 35 days of age.**

| BW | Control | CC | NCA | NCB | P value |
|---|---|---|---|---|---|
| Control | 2147.639 | | | | |
| Low | | 2115.493B | 2216.527 | 2170.515 | 0.1008 |
| Intermediate | | 2251.508A | 2153.38 | 2161.234 | 0.0708 |
| High | | 2231.462A | 2182.726 | 2131.332 | 0.1053 |
| SEM | | 10.3563 | 10.4372 | 10.6345 | |
| P value | | 0.0099 | 0.3982 | 0.6765 | |
| BWG | Control | CC | NCA | NCB | P value |
| Control | 2108.034 | | | | |
| Low | | 2075.888B | 2176.878 | 2130.822 | 0.101 |
| Intermediate | | 2211.876A | 2113.713 | 2121.506 | 0.0706 |
| High | | 2191.822A | 2143.059 | 2091.604 | 0.105 |
| SEM | | 10.7265 | 10.4523 | 10.6294 | |
| P value | | 0.01 | 0.3981 | 0.6763 | |
| FI | Control | CC | NCA | NCB | P value |
| Control | 3586.367 | | | | |
| Low | | 3579.563 | 3577.258 | 3662.793AB | 0.58 |
| Intermediate | | 3679.007 | 3552.885 | 3838.927A | 0.0128 |
| High | | 3685.459 | 3552.031 | 3512.373B | 0.1592 |
| SEM | | 40.7253 | 35.0561 | 34.0662 | |
| P value | | 0.4475 | 0.9536 | 0.0039 | |
| FCR | Control | CC | NCA | NCB | P value |
| Control | 1.704 | | | | |
| Low | | 1.725a | 1.643b | 1.719aB | 0.0425 |
| Intermediate | | 1.664b | 1.680b | 1.810aA | 0.0003 |
| High | | 1.681 | 1.657 | 1.679B | 0.7663 |
| SEM | | 0.0153 | 0.0184 | 0.0177 | |
| P value | | 0.2294 | 0.5825 | 0.0026 | |

BW–Body weight; BWG–Body weight gain; FI–Feed intake; FCR–Feed conversion ratio; a,b Means followed by different letters in the same line differ significantly Tukey test, P<0.05. A, B Means followed by different letters in the same collum differ significantly Tukey test, P<0.05.

reduced the steatosis score compared to the diet without choline supplementation (P<0.001), with a reduction observed in all three evaluated dosages compared to the control (diet origin; P<0.001).

In the interaction, steatosis showed that it is reduced with the supplementation of an additional source in the diet. However, better results were obtained with NCA and NCB compared to CC (P<0.001), as shown in Table 13. NCA showed a better result than NCB and CC even at the low dosage (P<0.001), consistently achieving better results at the intermediate and high dosages (P<0.001). A lower score was observed at the high dosage of NCA (P<0.001). The liver triglyceride content was higher with CC supplementation compared to NCA (P<0.001), but similar to NCB (P>0.05) at the intermediate dosage.

Findings from this trial suggest NCA outperformed other sources evaluated. Aside from superior performance, NCA was also able to provide satisfactory results at lower doses. To support use recommendations, economic estimates accounting for the effects of additive and dose on production were presented in Table 14 (Investment per kg of feed per phase and weighted average in days) and Table 15 (Relative Gross Income calculation).

**Table 8. Effect of choline supplementation on zootechnical parameters of broilers at 42 days of age.**

| Supplementation | BW | BWG | FI | FCR | LIVE % |
|---|---|---|---|---|---|
| Without | 2898.869 | 2859.263 | 4756.740 | 1.665ab | 94.737 |
| CC | 2938.525 | 2898.899 | 4954.271 | 1.710a | 95.029 |
| NCA | 2978.035 | 2938.374 | 4770.241 | 1.624b | 96.491 |
| NCB | 2916.491 | 2876.775 | 4884.642 | 1.699a | 96.491 |
| **Dosage[1]** | **BW42** | **BWG42** | **FI42** | **FCR42** | **LIVE42** |
| Control | 2898.869 | 2859.263 | 4756.740 | 1.665 | 94.737 |
| Low | 2950.548 | 2910.899 | 4856.164 | 1.670 | 95.029 |
| Intermediate | 2952.020 | 2912.345 | 4912.025 | 1.686 | 95.322 |
| High | 2930.483 | 2890.805 | 4840.964 | 1.677 | 97.661 |
| **Supplementation** | **0.3214** | **0.3216** | **0.2392** | **0.0432** | **0.6298** |
| **Dosage** | **0.7367** | **0.7374** | **0.7143** | **0.9541** | **0.249** |
| **Interaction** | **0.0166** | **0.0167** | **0.1926** | **0.0457** | **0.8955** |
| **SEM** | **46.343** | **46.3591** | **120.5903** | **0.391** | **1.8105** |
| **C.V. (%)** | **3.86** | **3.92** | **6.08** | **5.72** | **4.63** |

IBW–Initial body weight; BW–Body weight; BWG–Body weight gain; FI–Feed intake; FCR–Feed conversion ratio; LIVE–Liveability; Means followed by different letters in the same column differ significantly; Tukey test, P<0.05.

**Table 9. Effect of interaction between the supplementation sources of broilers at 42 days of age.**

| BW | Control | CC | NCA | NCB | P value |
|---|---|---|---|---|---|
| **Control** | 2898.869 | | | | |
| Low | | 2866.541bB | 3035.380aA | 2949.721ab | 0.0443 |
| Intermediate | | 2999.664a | 2943.436B | 2912.958 | 0.4127 |
| High | | 2949.368AB | 2955.287B | 2886.793 | 0.5159 |
| **SEM** | | 20.4835 | 19.5264 | 20.4273 | |
| **P value** | | 0.0326 | 0.0203 | 0.6306 | |
| **BWG** | **Control** | **CC** | **NCA** | **NCB** | **P value** |
| **Control** | 2859.263 | | | | |
| Low | | 2826.936b | 2995.730a | 2910.028ab | 0.0445 |
| Intermediate | | 2960.033 | 2903.77 | 2873.23 | 0.4122 |
| High | | 2909.728 | 2915.62 | 2847.065 | 0.5154 |
| SEM | | 16.7247 | 15.7255 | 17.9958 | |
| P value | | 0.1329 | 0.3204 | 0.6305 | |
| **FCR** | **Control** | **CC** | **NCA** | **NCB** | **P value** |
| **Control** | 1.665 | | | | |
| Low | | 1.7196 | 1.6 | 1.6911 | 0.0889 |
| Intermediate | | 1.682ab | 1.618b | 1.757a | 0.0405 |
| High | | 1.729 | 1.652 | 1.647 | 0.2577 |
| **SEM** | | 0.0213 | 0.0261 | 0.02196 | |
| **P value** | | 0.6611 | 0.6302 | 0.1446 | |

BW–Body weight; BWG–Body weight gain; FI–Feed intake; FCR–Feed conversion ratio; a,b Means followed by different letters in the same line differ significantly Tukey test, P<0.05. A, B Means followed by different letters in the same column differ significantly Tukey test, P<0.05.

**Table 10. Effect of choline supplementation on carcass characteristics of broilers at 42 days of age.**

| Supplementation | CY | BY | LY | FY | TY | DY |
|---|---|---|---|---|---|---|
| Without | 28.472 | 31.283a | 21.247 | 0.647 | 9.853b | 11.239 |
| CC | 28.467 | 31.388ab | 21.383 | 0.680 | 9.935ab | 11.440 |
| NCA | 29.057 | 31.259ab | 21.352 | 0.680 | 10.041ab | 11.473 |
| NCB | 28.936 | 30.707b | 21.548 | 0.646 | 10.133a | 11.542 |
| **Dosage[1]** | | | | | | |
| Control | 28.472 | 31.283 | 21.247 | 0.647 | 9.853b | 11.239 |
| Low | 29.109 | 31.213 | 21.606 | 0.689 | 10.150a | 11.510 |
| Intermediate | 28.676 | 31.065 | 21.273 | 0.665 | 10.048ab | 11.484 |
| High | 28.674 | 31.077 | 21.405 | 0.652 | 9.910ab | 11.461 |
| **Supplementation** | **0.0544** | **0.0036** | **0.5018** | **0.623** | **0.0251** | **0.1225** |
| **Dosage** | **0.1278** | **0.07729** | **0.1832** | **0.6674** | **0.0065** | **0.1938** |
| **Interaction** | **0.5198** | **0.02** | **0.031** | **0.1141** | **0.5095** | **0.1** |
| **SEM** | **0.291** | **0.261** | **0.201** | **0.04** | **0.437** | **0.499** |
| **C.V. (%)** | **5.53** | **4.25** | **5.15** | **32.54** | **5.3** | **5.29** |

CY–Carcass yield %; BY–Breast yield %); LY–Leg yield %; FY—Fat yield %; TY–Thigh yield %; DY–Drumstick yield %; Means followed by different letters in the same column differ significantly; Tukey test, P<0.05.

Combined analysis of liveability (*i.e.*, the final number of birds per treatment), performance, carcass yield, and economic data, which can be adjusted according to natural variations, revealed relative gross margin fluctuations across different scenarios (Table 14).

Using to the Control treatment as a reference, the relative gross income derived from breast, thigh, drumstick, and whole chicken sales is 100%. The same applies to the remaining treatments in the same order. Hence, 100 is a used benchmark to estimate the delta between treatments.

Based on comparative analysis between the control and remaining treatments, NCA fed at 100 g/t yields the highest relative gross income (107.42, *i.e.*, 7.42 p.p higher the than control).

**Table 11. Effect of interaction between the supplementation sources of broilers at 42 days of age.**

| Breast Yield % | Control | CC | NCA | NCB | P value |
|---|---|---|---|---|---|
| Control | 31.283 | | | | |
| Low | | 31.184 | 31.627 | 30.827 | 0.0656 |
| Intermediate | | 31.251 | 30.962 | 30.98 | 0.6401 |
| High | | 31.728a | 31.188a | 30.313b | 0.0002 |
| SEM | | 0.188 | 0.154 | 0.172 | |
| P value | | 0.2233 | 0.1429 | 0.1257 | |
| Thighdrums Yield % | Control | CC | NCA | NCB | P value |
| Control | 21.247 | | | | |
| Low | | 21.482ab | 21.317b | 22.017aA | 0.0383 |
| Intermediate | | 21.428 | 21.394 | 20.998B | 0.249 |
| High | | 21.242 | 21.343 | 21.628AB | 0.3752 |
| SEM | | 0.131 | 0.101 | 0.164 | |
| P value | | 0.6804 | 0.9626 | 0.0017 | |

a,b Means followed by different letters in the same line differ significantly Tukey test, P<0.05. A, B Means followed by different letters in the same collum differ significantly Tukey test, P<0.05.

**Table 12. Effects of treatments on liver parameters of broilers at 42 days of age.**

| Supplementation | LC | LT | LL | SS |
|---|---|---|---|---|
| Without | 187.254 | 240.118 | 2.207 | 3.311a |
| CC | 162.090 | 305.788 | 2.719 | 2.970b |
| NCA | 144.885 | 215.012 | 3.151 | 2.259d |
| NCB | 152.819 | 238.510 | 2.659 | 2.733c |
| **Dosage[1]** | | | | |
| Control | 187.254a | 240.118 | 2.207b | 3.311a |
| Low | 159.284ab | 241.614 | 3.295a | 2.867b |
| Intermediate | 131.476b | 261.172 | 2.578ab | 2.741b |
| High | 169.034ab | 256.524 | 2.656ab | 2.356c |
| **Supplementation** | **0.3138** | **0.052** | **0.1442** | **<0.001** |
| **Dosage** | **0.0591** | **0.9167** | **0.0394** | **<0.001** |
| **Interaction** | **0.6476** | **0.0395** | **0.0623** | **<0.001** |
| **SEM** | 20.006 | 39.204 | 0.370 | 0.0909 |
| **C.V. (%)** | 25.54 | 31.14 | 26.63 | 22.44 |

LC–Liver cholesterol (mg/dL); LT–Liver triglycerides (mg/dL); LL–Liver lipids (%); SS–Steatosis score; Means followed by different letters in the same column differ significantly; Tukey test, $P<0.05$.

Margins derived from NCA at 100 g/ton are also higher relative to the three different doses of Choline Chloride or the NCB product (400 g/t Choline Chloride vs. 100 g/t NCA; 10.54 points; 800 g/t Choline Chloride vs. 200 g/t NCA, 4.06 p.p.; 1,200 g/t Choline Chloride vs. 300 g/t NCA, 4.40 points). NCA fed at 200 g/t, or 300 g/t outperformed the control and Choline Chloride fed at three different doses.

The NCB product was economically superior to the control treatment when fed at 100 g/t and 300 g/t doses, However, 0.6 p.p. losses were estimated at the 200 g/t dose. The NCB product outperformed Choline Chloride fed at 400g/t. In contrast, when fed at 200 g/t, the economic performance of the NCB product was poorer (4 to 5 points less) compared to Choline

**Table 13. Effect of interaction between supplementation sources of broilers at 42 days of age.**

| Steatosis | Control | CC | NCA | NCB | P value |
|---|---|---|---|---|---|
| Control | 3.311 | | | | |
| Low | | 3.288aA | 2.488cB | 2.82b | <0.001 |
| Intermediate | | 3.066aA | 2.355cA | 2.80a | <0.001 |
| High | | 2.555aB | 1.933bA | 2.577a | <0.001 |
| SEM | | 0.632 | 0.472 | 0.516 | |
| P value | | <0.001 | <0.001 | 0.1112 | |
| Triglycerides | Control | CC | NCA | NCB | P value |
| Control | 240.118 | | | | |
| Low | | 300.428 | 242.694 | 181.718 | 0.1184 |
| Intermediate | | 361.959a | 159.404b | 262.154ab | 0.004 |
| High | | 254.976 | 242.938 | 271.657 | 0.874 |
| SEM | | 18.383 | 15.010 | 17.652 | |
| P value | | 0.1708 | 0.2375 | 0.2209 | |

a,b Means followed by different letters in the same line differ significantly Tukey test, $P<0.05$. A, B Means followed by different letters in the same column differ significantly Tukey test, $P<0.05$.

**Table 14. Investment per kg of feed per treatment (BRL/kg of feed/treatment) at the different stages of production in broilers aged 1 to 42 days.**

| Age | Control | Choline Chloride 60% | | | NCA | | | NCB | | |
|---|---|---|---|---|---|---|---|---|---|---|
| | | 400 g/t | 800 g/t | 1,200 g/t | 100 g/t | 200 g/t | 300 g/t | 100 g/t | 200 g/t | 300 g/t |
| 1–21 d | 2.283 | 2.288 | 2.293 | 2.302 | 2.286 | 2.288 | 2.291 | 2.286 | 2.289 | 2.292 |
| 22–35 d | 2.267 | 2.272 | 2.278 | 2.283 | 2.270 | 2.272 | 2.275 | 2.270 | 2.273 | 2.277 |
| 36–42 d | 2.224 | 2.231 | 2.236 | 2.240 | 2.228 | 2.231 | 2.234 | 2.229 | 2.232 | 2.235 |
| 1–42 d | 2.268 | 2.273 | 2.278 | 2.286 | 2.271 | 2.273 | 2.276 | 2.271 | 2.274 | 2.278 |

Price estimates based on values listed in feed Tables, July 2022.

**Table 15. Economic assessment.**

| | Control | Choline Chloride, g/t | | | NCA g/t | | | NCB, g/t | | |
|---|---|---|---|---|---|---|---|---|---|---|
| | | 400 | 800 | 1,200 | 100 | 200 | 300 | 100 | 200 | 300 |
| Liveability, % | 94.74 | 93.86 | 94.74 | 96.49 | 94.74 | 95.61 | 99.12 | 96.49 | 95.61 | 97.37 |
| Weight gain, kg | 2.86 | 2.83 | 2.96 | 2.91 | 3.00 | 2.90 | 2.92 | 2.91 | 2.87 | 2.85 |
| Feed conversion ratio, kg/kg | 1.67 | 1.72 | 1.68 | 1.73 | 1.60 | 1.62 | 1.65 | 1.69 | 1.76 | 1.65 |
| Production Factor | 386.21 | 367.30 | 397.44 | 386.40 | 422.34 | 408.04 | 417.02 | 395.59 | 371.63 | 400.03 |
| Birds Final Adjusted | 142.00 | 141.00 | 142.00 | 145.00 | 142.00 | 143.00 | 149.00 | 145.00 | 143.00 | 146.00 |
| Feed Intake, kg/treat | 675.46 | 685.04 | 707.76 | 727.93 | 680.45 | 672.14 | 717.96 | 713.14 | 722.38 | 683.89 |
| Cost of diet, BRL/kg | BRL 2.27 | BRL 2.27 | BRL 2.27 | BRL 2.28 | BRL 2.27 | BRL 2.27 | BRL 2.27 | BRL 2.27 | BRL 2.28 | BRL 2.28 |
| Cost of feed, BRL/kg | BRL 1,531.84 | BRL 1,555.31 | BRL 1,608.81 | BRL 1,657.90 | BRL 1,543.97 | BRL 1,525.76 | BRL 1,630.53 | BRL 1,620.70 | BRL 1,645.00 | BRL 1,560.54 |
| Broiler production, kg/treat | 411.64 | 404.18 | 425.95 | 427.66 | 431.02 | 420.91 | 440.34 | 427.71 | 416.55 | 421.47 |
| Cost of Broiler production, BRL/kg | BRL 3.72 | BRL 3.85 | BRL 3.78 | BRL 3.88 | BRL 3.58 | BRL 3.62 | BRL 3.70 | BRL 3.79 | BRL 3.95 | BRL 3.70 |
| Price of broiler, BRL/kg | BRL 7.00 | BRL 7.00 | BRL 7.00 | BRL 7.00 | BRL 7.00 | BRL 7.00 | BRL 7.00 | BRL 7.00 | BRL 7.00 | BRL 7.00 |
| Breast production, kg/treat | 128.76 | 126.02 | 133.11 | 135.70 | 136.33 | 130.31 | 137.34 | 131.86 | 129.05 | 127.75 |
| Cost/Breast, BRL/kg | BRL 11.90 | BRL 12.34 | BRL 12.09 | BRL 12.22 | BRL 11.33 | BRL 11.71 | BRL 11.87 | BRL 12.29 | BRL 12.75 | BRL 12.22 |
| Price of breast, BRL/kg | BRL 13.00 | BRL 13.00 | BRL 13.00 | BRL 13.00 | BRL 13.00 | BRL 13.00 | BRL 13.00 | BRL 13.00 | BRL 13.00 | BRL 13.00 |
| Thigh production, kg/treat | 40.55 | 40.54 | 42.38 | 42.00 | 43.79 | 42.81 | 43.11 | 43.84 | 41.78 | 42.61 |
| Cost/Thigh, BRL/kg | BRL 37.78 | BRL 38.37 | BRL 37.96 | BRL 39.48 | BRL 35.26 | BRL 35.64 | BRL 37.82 | BRL 36.97 | BRL 39.37 | BRL 36.62 |
| Price of the thigh, BRL/kg | BRL 12.00 | BRL 12.00 | BRL 12.00 | BRL 12.00 | BRL 12.00 | BRL 12.00 | BRL 12.00 | BRL 12.00 | BRL 12.00 | BRL 12.00 |
| Drumstick, kg/treat | 46.27 | 46.12 | 49.03 | 48.75 | 49.74 | 48.19 | 50.33 | 49.53 | 47.86 | 48.72 |
| Cost/Drumstick, BRL/kg | BRL 33.11 | BRL 33.73 | BRL 32.81 | BRL 34.01 | BRL 31.04 | BRL 31.66 | BRL 32.40 | BRL 32.72 | BRL 34.37 | BRL 32.03 |
| Price of drumstick, BRL/kg | BRL 12.00 | BRL 12.00 | BRL 12.00 | BRL 12.00 | BRL 12.00 | BRL 12.00 | BRL 12.00 | BRL 12.00 | BRL 12.00 | BRL 12.00 |
| Gross income, BRL (Breast + Thigh +Drumstick) | BRL 5,597.15 | BRL 5,507.47 | BRL 5,809.00 | BRL 5,846.65 | BRL 5,911.88 | BRL 5,732.48 | BRL 5,989.08 | BRL 5,828.61 | BRL 5,669.21 | BRL 5,707.02 |
| Gross margin, BRL (Gross Income—Cost of feed) | BRL 4,065.30 | BRL 3,952.15 | BRL 4,200.19 | BRL 4,188.74 | BRL 4,367.91 | BRL 4,206.72 | BRL 4,358.55 | BRL 4,207.90 | BRL 4,024.21 | BRL 4,146.48 |
| Relative Gross margin, % | 100.00 | 97.17 | 103.23 | 102.89 | 107.42 | 103.42 | 107.13 | 103.53 | 99.04 | 102.07 |
| | | 100.00 | 106.23 | 105.89 | 110.54 | 106.43 | 110.24 | 106.55 | 101.92 | 105.04 |
| | | | 100.00 | 99.68 | 104.06 | 100.19 | 103.78 | 100.30 | 95.95 | 98.88 |
| | | | | 100.00 | 104.40 | 100.52 | 104.11 | 100.62 | 96.25 | 99.20 |
| | | | | | 100.00 | 96.28 | 103.58 | 96.64 | 95.66 | 103.06 |
| | | | | | | 100.00 | 103.58 | 100.10 | 95.76 | 98.69 |
| | | | | | | | 100.00 | 96.64 | 92.45 | 95.28 |
| | | | | | | | | 100.00 | 95.66 | 98.59 |
| | | | | | | | | | 100.00 | 103.06 |

Color scale: color changes indicate a 10% change in value. Red: lowest value. Dark green: highest value. Yellow, orange, light green: 10% value difference intervals.
Benchmark: 100.

Chloride fed at 800 g/t or 1,200 g/t (Table 10; highlights in red). Assessment estimates suggest NCA was more efficient than the control treatment, Choline Chloride or NCB at doses used in this trial.

## Discussion

During the total period of this trial (1–42 days), statistical differences were observed for the performance variables evaluated by Tukey analysis. Furthermore, employing orthogonal contrasts, it was evident that NCA exhibited superior performance compared to the NCB product. This could be due to the higher bioavailability or better absorption of NCA in the body. Specifically, NCA demonstrated notable enhancements in terms of body weight (BW) with an increment of 61.54 g/bird (P = 0.0143), body weight gain (BWG) with an increment of 61.59 g/bird (P = 0.0142), and feed conversion ratio (FCR) with an improvement of 7.6 points (P = 0.0190). These improvements indicate that NCA is more effective in promoting growth and improving feed efficiency in birds. Furthermore, birds fed with NCA displayed a trend of lower feed intake by 114.4 g/bird, although this difference was not statistically significant. Suggest that NCA might be more satiating or more efficiently utilized by the birds, leading to lower feed intake.

This aligns with a study by [18] that suggested herbal choline supplementation at 0.350 and 0.500 kg/ton of feed can effectively substitute synthetic choline chloride-60%, typically used at 1 kg/ton of broiler feed. Indicates that natural sources of choline can be just as effective, if not more so, than synthetic sources. Similarly, [12] evaluated the use of a vegetal source of choline in five levels of supplementation as a replacement of choline chloride at 60% and did not find any difference between the sources on the performance parameters, but they concluded that the use of 100 mg.kg-1 of a vegetal source of choline to replace the use of choline chloride in corn-soybean meal diets for broilers from 1 to 42 days of age, further supports the idea that natural sources of choline can effectively replace synthetic sources without compromising performance. Likewise, [19] reported comparable body weight outcomes for both natural and synthetic choline-supplemented groups, suggests that the source of choline, whether natural or synthetic, does not significantly affect body weight outcomes. Correspondingly, [5] demonstrated that the NCA at 400 g/ton replaced the function of 1 kg/ton of synthetic choline (choline chloride 60%) in broilers, showing the potential to mimic biological activities of synthetic choline through the restoration of negative effects caused by a choline-deficient diet, so indicates that NCA can be used at lower doses than synthetic choline while still providing the same benefits. On the other hand, [20] investigated the effects of replacing synthetic choline with a natural choline source and demonstrated no significative difference on zootechnical performance and nutrition utilization and carcass characteristics, suggests that the source of choline does not significantly affect these parameters. However, the current results clearly demonstrates the importance of choline supplementation and additionally, the results demonstrate the superiority of natural sources over synthetic ones, with significant results, especially with NCA, and because of that, using natural sources of choline, such as NCA, can provide superior results in terms of performance and economic efficiency.

Exploring carcass characteristics, the study revealed that at 200g/t, NCA led to an increase in leg weight, while at 300g/t, it caused a decrease in thigh weight and an increase in breast weight compared to NCB at 300g/t, suggests that NCA dosage can influence the distribution of muscle mass in broilers. Additionally, NCA at 200g/t exhibited a higher carcass yield compared to Choline chloride at 800g/t. These findings were corroborated by [4], which demonstrated the advantages of herbal choline supplementation at 0.5 kg/t of feed in terms of enhanced liver protection, carcass traits, and economic viability in broiler production, supports the potential benefits of using herbal alternatives to traditional supplements. Moreover, a

comparative analysis was conducted among NCA, choline chloride, and NCB. Significant distinctions emerged between NCA versus Choline chloride and NCA versus NCB in terms of controlling fatty liver. NCA exhibited greater efficacy in mitigating fatty liver incidence in broilers compared to choline chloride and NCB, showing a dose-dependent effect, and because of that, that NCA can be a promising supplement for liver health in broilers. In this study, NCA at 200 g/t or 300 g/t outperformed the Control and Choline Chloride at various doses. In a similar vein, [21] reported reduced fatty liver incidence in broilers through herbal choline feeding. NCA contains certain compounds with proven hepatoprotective effects, such as polyphenols & curcuminoids. Therefore, findings from this trial, particularly the significant reduction in fatty liver development, may be attributed to curcumin and catechin contained in the natural source of choline employed. Studies have shown that curcumin supplementation can improve the growth performance of broiler chickens [22–25]. Moreover, the phytoconstituents (polyphenols and curcuminoids) present in NCA modulates the liver genes accounts for fat metabolism/catabolism and lipogenesis [26]. This might enhance the energy availability at the muscular level which inturn enhanced fat utilization. Because of that, the NCA outperformed CC and NCB, as shown by better animal performance parameters and best carcass yields. In summary, the study suggests that NCA, particularly at doses of 200 g/t or 300 g/t, outperforms both the Control and Choline Chloride at various doses in terms of carcass characteristics and liver health in broilers.

As anticipated before, combined analysis of performance data and economic estimates suggest a beneficial effect of NCA compared to remaining treatments, and that NCA given at the 100 g/t dose yields the highest relative gross margin, in this terms, NCA is not Only effective in imporving broiler performance but also costo-effective, and more, NCS is more effective than NCB and CC in promoting growth and yield in broilers. Because of these effects, economic assessment findings estimate to support the superiority of NCA relative to NCB and CC in this trial in terms of animal performance as well as carcass and cut yield in broilers aged 1 to 42 days, particularly when fed at 100 g/t. In this study's conditions, the dosage of 100g/ton probably was enough to attend to the broilers' requirements due to the low oxidative stress conditions and the use of a corn-soy-based diet. In field conditions with challenges such as low quality of ingredients, heat stress, and high density of animals, higher doses of the NCA evaluated in this study (200g/t or 300g/t) can be recommended. From a practical standpoint (i.e., product storage and feed logistics) the lower dose requirement is another advantage of NCA relative to Choline Chloride, the commercial doses recommended of CC (400, 800, or 1,200 g/t) can involve several space-related and logistics concerns. Therefore, the incorporation of NCA at a dosage of 100g/t in broiler feed could serve as a viable substitute for choline chloride and NCB.

## Conclusion

Natural Choline A is a superior source of supplemental choline, offering enhanced hepatoprotective effects and improved animal performance and carcass quality. It has greater economic efficiency compared to Chloride Choline and Natural Choline B. For optimal results, a dose of 100 g/t of Natural Choline A is recommended under low oxidative stress conditions. In conditions of higher heat stress, doses of 200g/t or 300g/t of Natural Choline A are recommended for broilers from 1 to 42 days of age.

## Supporting information

**S1 Data.**
(XLSX)

## Author Contributions

**Conceptualization:** Matheus Ramalho de Lima, Fernando Guilherme Perazzo Costa, Andreia D. C. Vilas Boas, Ana Louise Toledo, Sigfrido Lopez Ferrer, Saravanakumar Marimuthu.

**Data curation:** Matheus Ramalho de Lima, Fernando Guilherme Perazzo Costa, Andreia D. C. Vilas Boas, Ana Louise Toledo, Sigfrido Lopez Ferrer, Saravanakumar Marimuthu.

**Formal analysis:** Matheus Ramalho de Lima, Fernando Guilherme Perazzo Costa.

**Funding acquisition:** Matheus Ramalho de Lima, Fernando Guilherme Perazzo Costa, Andreia D. C. Vilas Boas, Ana Louise Toledo, Sigfrido Lopez Ferrer, Saravanakumar Marimuthu.

**Investigation:** Matheus Ramalho de Lima, Isabelle Naemi Kaneko, Adiel Vieira de Lima, Lucas Nunes de Melo, Mario Cesar de Lima, Anna Neusa Eduarda Ferreira de Brito, Fernando Guilherme Perazzo Costa, Andreia D. C. Vilas Boas, Sigfrido Lopez Ferrer, Saravanakumar Marimuthu.

**Methodology:** Matheus Ramalho de Lima, Fernando Guilherme Perazzo Costa, Andreia D. C. Vilas Boas, Sigfrido Lopez Ferrer, Saravanakumar Marimuthu.

**Project administration:** Matheus Ramalho de Lima, Fernando Guilherme Perazzo Costa, Andreia D. C. Vilas Boas, Ana Louise Toledo.

**Resources:** Matheus Ramalho de Lima, Fernando Guilherme Perazzo Costa, Sigfrido Lopez Ferrer.

**Software:** Matheus Ramalho de Lima, Fernando Guilherme Perazzo Costa, Saravanakumar Marimuthu.

**Supervision:** Matheus Ramalho de Lima, Isabelle Naemi Kaneko, Fernando Guilherme Perazzo Costa, Andreia D. C. Vilas Boas.

**Validation:** Matheus Ramalho de Lima, Isabelle Naemi Kaneko, Adiel Vieira de Lima, Fernando Guilherme Perazzo Costa, Andreia D. C. Vilas Boas, Ana Louise Toledo, Sigfrido Lopez Ferrer, Saravanakumar Marimuthu.

**Visualization:** Matheus Ramalho de Lima, Adiel Vieira de Lima, Lucas Nunes de Melo, Mario Cesar de Lima, Anna Neusa Eduarda Ferreira de Brito, Fernando Guilherme Perazzo Costa, Ana Louise Toledo, Saravanakumar Marimuthu.

**Writing – original draft:** Matheus Ramalho de Lima, Fernando Guilherme Perazzo Costa.

**Writing – review & editing:** Matheus Ramalho de Lima, Fernando Guilherme Perazzo Costa, Saravanakumar Marimuthu.

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
