## [Decision Letter · Decision Letter 0]

23 Jan 2024

PONE-D-23-39073Choline through diet or supplementation? Synthetic or natural source supplementation? Improving performance, reducing steatosis, and economic viability of broiler chickens from 1 to 42 days.PLOS ONE

Dear Dr. Ramalho de Lima,

Thank you for submitting your manuscript to PLOS ONE. After careful consideration, we feel that it has merit but does not fully meet PLOS ONE’s publication criteria as it currently stands. Therefore, we invite you to submit a revised version of the manuscript that addresses the points raised during the review process. Please submit your revised manuscript by Mar 08 2024 11:59PM. If you will need more time than this to complete your revisions, please reply to this message or contact the journal office at plosone@plos.org. Please include the following items when submitting your revised manuscript:A rebuttal letter that responds to each point raised by the academic editor and reviewer(s). You should upload this letter as a separate file labeled 'Response to Reviewers'.A marked-up copy of your manuscript that highlights changes made to the original version. You should upload this as a separate file labeled 'Revised Manuscript with Track Changes'.An unmarked version of your revised paper without tracked changes. You should upload this as a separate file labeled 'Manuscript'.

We look forward to receiving your revised manuscript.

Kind regards,

Ewa Tomaszewska, DVM Ph.D

Academic Editor

PLOS ONE

Journal Requirements:

Reviewers' comments:

Reviewer's Responses to Questions

**Comments to the Author**

1. Is the manuscript technically sound, and do the data support the conclusions?

Reviewer #1: Partly

2. Has the statistical analysis been performed appropriately and rigorously? 

Reviewer #1: No

3. Have the authors made all data underlying the findings in their manuscript fully available?

Reviewer #1: No

4. Is the manuscript presented in an intelligible fashion and written in standard English?

Reviewer #1: Yes

5. Review Comments to the Author

Reviewer #1: Dear Authors,

strictly follow all my corrections/comments in attached file, while revising the manuscript make comments viz author's response for better understanding and future suggestion.

Thank You!

6. PLOS authors have the option to publish the peer review history of their article (what does this mean?). If published, this will include your full peer review and any attached files.

Reviewer #1: No

---

## [Author Response · Author response to Decision Letter 0]

1 Feb 2024

Dear Reviewers,

Thank you for reviewing our manuscript. 

We have made and accepted all the suggested recommendations, so we list them point by point:

• The title has been adjusted, displaying more concise information. 

• The abstract has been adjusted. Information corrected and added regarding the experimental design and data analysis. 

• The hypothesis was added to the introduction as requested. 

• In the materials and methods, the recommended data were added. I highlight the data on the animals and environmental conditions, the data analysis and mathematical model, as well as the sexing of the animals in the hatchery. 

• The variables were adjusted as requested. Including more information on how they were obtained. • In the statistical analysis, the mathematical model was added. 

• In the results, the standard error of the mean was inserted in all result tables. 

• More information was added to the discussion, improving the fluidity and explanatory information of the effects obtained in the study. 

• The conclusion was edited, adjusting as requested.

---

## [Decision Letter · Decision Letter 1]

13 Feb 2024

Choline supplementation: Impact on Broiler Chicken Performance, Steatosis, and Economic Viability from from 1 to 42 days

PONE-D-23-39073R1

Dear Dr. Matheus Ramalho de Lima,

We’re pleased to inform you that your manuscript has been judged scientifically suitable for publication and will be formally accepted for publication once it meets all outstanding technical requirements.

Kind regards,

Ewa Tomaszewska, DVM Ph.D

Academic Editor

PLOS ONE

Additional Editor Comments (optional):

Reviewers' comments:

Reviewer's Responses to Questions

**Comments to the Author**

1. If the authors have adequately addressed your comments raised in a previous round of review and you feel that this manuscript is now acceptable for publication, you may indicate that here to bypass the “Comments to the Author” section, enter your conflict of interest statement in the “Confidential to Editor” section, and submit your "Accept" recommendation.

Reviewer #1: All comments have been addressed

2. Is the manuscript technically sound, and do the data support the conclusions?

Reviewer #1: Yes

3. Has the statistical analysis been performed appropriately and rigorously? 

Reviewer #1: Yes

4. Have the authors made all data underlying the findings in their manuscript fully available?

Reviewer #1: Yes

5. Is the manuscript presented in an intelligible fashion and written in standard English?

Reviewer #1: Yes

6. Review Comments to the Author

Reviewer #1: Dear Authors,

I have completed my evaluation, and all my concerns were satisfactorily addressed and now manuscript is acceptable form my side.

Best,

7. PLOS authors have the option to publish the peer review history of their article (what does this mean?). If published, this will include your full peer review and any attached files.

Reviewer #1: **Yes: **Sohail Ahmad

---

## [Editor Report · Acceptance letter]

5 Mar 2024

PONE-D-23-39073R1 

PLOS ONE

Dear Dr. Ramalho de Lima, 

I'm pleased to inform you that your manuscript has been deemed suitable for publication in PLOS ONE. Congratulations! Your manuscript is now being handed over to our production team.

Kind regards, 

on behalf of

Professor Ewa Tomaszewska 

Academic Editor

PLOS ONE